# Depletion of R270C Mutant p53 in Osteosarcoma Attenuates Cell Growth but Does Not Prevent Invasion and Metastasis In Vivo

**DOI:** 10.3390/cells11223614

**Published:** 2022-11-15

**Authors:** Takatsune Shimizu, Eiji Sugihara, Hideyuki Takeshima, Hiroyuki Nobusue, Rui Yamaguchi, Sayaka Yamaguchi-Iwai, Yumi Fukuchi, Toshikazu Ushijima, Akihiro Muto, Hideyuki Saya

**Affiliations:** 1Department of Pathophysiology, School of Pharmacy and Pharmaceutical Sciences, Hoshi University, 2-4-41 Ebara, Shinagawa-ku, Tokyo 142-8501, Japan; 2Division of Gene Regulation, Institute for Advanced Medical Research, Keio University School of Medicine, 35 Shinanomachi, Shinjuku-ku, Tokyo 160-8582, Japan; 3Open Facility Center, Research Promotion Headquarters, Fujita Health University, Toyoake 470-1192, Japan; 4Division of Gene Regulation, Cancer Center, Research Promotion Headquarters, Fujita Health University, Toyoake 470-1192, Japan; 5Department of Epigenomics, Life Science Tokyo Advanced Research Center, Hoshi University, 2-4-41 Ebara, Shinagawa-ku, Tokyo 142-8501, Japan; 6Division of Cancer Systems Biology, Aichi Cancer Center Research Institute, Nagoya 464-8681, Japan; 7Division of Cancer Informatics, Nagoya University Graduate School of Medicine, Nagoya 466-8550, Japan; 8Department of Orthopedic Surgery, Keio University School of Medicine, 35 Shinanomachi, Shinjuku-ku, Tokyo 160-8582, Japan

**Keywords:** osteosarcoma, p53, mutant p53, metastasis

## Abstract

Novel therapeutic targets are needed to better treat osteosarcoma, which is the most common bone malignancy. We previously developed mouse osteosarcoma cells, designated AX (accelerated bone formation) cells from bone marrow stromal cells. AX cells harbor both wild-type and mutant forms of p53 (R270C in the DNA-binding domain, which is equivalent to human R273C). In this study, we showed that mutant p53 did not suppress the transcriptional activation function of wild-type p53 in AX cells. Notably, AXT cells, which are cells derived from tumors originating from AX cells, lost wild-type p53 expression, were devoid of the intact transcription activation function, and were resistant to doxorubicin. ChIP-seq analyses revealed that this mutant form of p53 bound to chromatin in the vicinity of the transcription start sites of various genes but exhibited a different binding profile from wild-type p53. The knockout of mutant p53 in AX and AXT cells by CRISPR–Cas9 attenuated tumor growth but did not affect the invasion of these cells. In addition, depletion of mutant p53 did not prevent metastasis in vivo. Therefore, the therapeutic potency targeting R270C (equivalent to human R273C) mutant p53 is limited in osteosarcoma. However, considering the heterogeneous nature of osteosarcoma, it is important to further evaluate the biological and clinical significance of mutant p53 in various cases.

## 1. Introduction

Osteosarcoma is the most common type of primary bone tumor in childhood and adolescence. The tumor is highly malignant, and although recent treatment advances that combine surgery and chemotherapy have improved prognosis, long-term survival is not achieved in ~30% of patients, mainly due to uncontrollable metastasis [1,2,3,4]. Accordingly, novel treatment options for overcoming therapeutic resistance are urgently required.

Previously, we developed a mouse model of osteosarcoma that used bone marrow stromal cells derived from *Ink4a/Arf* (*Cdkn2a*)-null mice overexpressing c-MYC [5]. When these highly tumorigenic cells, designated AX cells, are inoculated into C57BL/6 syngeneic mice, they rapidly form lethal tumors with metastatic lesions that mimic human osteoblastic osteosarcoma [6,7,8,9,10,11]. Based on exome-sequencing analysis, we found that AX cells harbor a heterozygous mutant form of p53. Moreover, AXT (AX-derived tumor) cells, which are cells derived from tumors originating from AX cells, exhibit loss of heterozygosity having lost a wild-type p53 allele.

Human TP53 is the most frequently mutated gene in human cancers; mutations in TP53 are found in almost half of malignant tumors [12,13,14]. The presence of mutated TP53 in osteosarcoma was previously thought to be relatively low; for example, the incidence in sporadic osteosarcoma has been reported to be less than 10% [15,16]. However, the use of conventional methods such as limited exon sequencing or immunohistochemistry to detect the accumulation of mutated p53 (which has a long half-life) might underestimate alterations in p53. The use of recent techniques including whole genome sequencing analyses has revealed that the alterations in p53 and the p53 pathway in osteosarcoma are more frequent. For example, exon sequencing of 196 osteosarcoma specimens detected 19.4% single-nucleotide variants in exon 4–10 of the p53 gene [17]. A whole genome sequencing study of 20 osteosarcomas from 19 patients demonstrated that all tumor samples harbor some abnormality in the p53 pathway [18]. In that study, 95% of the samples had either sequence mutations including three missense or frameshift mutations in the DNA-binding domain or structural variants in the p53 gene. A recent genome-wide study of 30 osteosarcoma samples from 23 patients detected structural variations and somatic nucleotide variants of p53 in 74% of the specimens [19]. In addition, the germline mutation of the p53 gene in Li–Fraumeni syndrome predisposes individuals (approximately 12%) to osteosarcoma [20,21,22], adding further evidence to the higher prevalence of mutated p53 in osteosarcoma. Animal genetic studies also indicate the importance of p53 dysfunction in the development of osteosarcoma [23,24,25,26]. Mouse osteosarcoma originates from lineage-committed immature osteoblasts or mesenchymal stem cells that have a combined loss of p53 and Rb [27,28]. Collectively, these findings suggest that p53 mutations are among the major oncogenic stimuli in osteosarcoma initiation. Mutant p53 is suggested to be a potential therapeutic target in malignant tumors, including osteosarcoma [29,30,31,32,33]. However, whether the mutant form of p53 already present in osteosarcoma is involved in maintaining osteosarcoma remains to be elucidated. In particular, the roles of mutant p53 in the progression of both primary lesions and metastasis in osteosarcoma have not yet been fully analyzed in vivo.

In this study, we examined the in vitro and in vivo role of the R270C form of mutant p53, which is equivalent to the commonly occurring human R273C mutant, using our previously developed syngeneic mouse model of osteosarcoma. We also discussed the feasibility of the R270C p53 mutant as a therapeutic target for osteosarcoma.

## 2. Materials and Methods

### 2.1. Cell Culture

Mouse osteosarcoma AX, AXT, and AO (adipo-, osteo-differentiation) cells were established by overexpressing c-MYC in bone marrow stromal cells derived from *Ink4a/Arf*-null mice [5]. The cells were cultured in IMDM (Nacalai Tesque, Kyoto, Japan) supplemented with 10% FBS for AX and AXT cells or with 20% FBS for AO cells under 5% CO_2_ at 37 °C [6,10,11].

### 2.2. Cell Proliferation Assay

The cells were treated with trypsin, collected, and washed with a serum-free medium. AX or AXT cells, including the p53-knockout AX or AXT cells, were then transferred to 96-well cell culture plates (1 × 10^3^ cells per well in 50 µL IMDM supplemented with 10% FBS). The cells were incubated for 1 h before the addition of 50 µL of the corresponding medium supplemented with doxorubicin at twice the desired final concentrations. After incubation, cell viability was measured using a Cell Titer Glo assay kit (Promega, Madison, WI, USA). Assays were performed at least in triplicate, and data are expressed as the means ± SD relative (fold change) to the corresponding control value for the cells incubated in the absence of doxorubicin.

### 2.3. Invasion Assay

Invasive ability was evaluated using a CytoSelect 24-well cell invasion assay kit (Cell Biolabs, San Diego, CA, USA). Briefly, 3.5 × 10^5^ AX cells or 1.5 × 10^5^ AXT cells, including the p53-knockout AX or AXT cells, were suspended in a serum-free IMDM, seeded into a Matrigel-coated upper chamber, and allowed to invade the basement membrane matrix for 26 h (AX cells) or 24 h (AXT cells). The IMDM supplemented with 10% FBS was added in the lower chamber. After incubation, noninvasive cells on the inside of inserts were removed, and the invasive cells that passed through the pores were stained and counted in each of the three different fields. The assays were performed in triplicate.

### 2.4. Reverse Transcription (RT) and Real-Time PCR Analysis

Total RNA extraction, RT, and real-time PCR analyses were performed using a NucleoSpin RNA kit and a PrimeScript reverse transcriptase (Takara, Shiga, Japan) [6,8,10]. The sequences of PCR primers are provided in Table 1. To prepare samples from mouse in vivo studies, the left lung was suspended in a lysis buffer and disrupted using a BioMasher (Nippi, Tokyo, Japan). To evaluate the circulating tumor cells, total RNA was extracted from 200 μL blood with a NucleoSpin RNA blood kit (Takara). Since GFP was expressed in the AX and AXT cells, tumor cells were quantitated based on the level of *Gfp* mRNA expression relative to *Actb* mRNA expression.

### 2.5. Detection of the Missense Mutation of p53

Total RNA was collected from AX, AXT, or AO cells, and cDNA was synthesized. The region including the mutation was amplified with forward primer 5′-CAAGTACATGTGTAATAGCTCCT-3′ and reverse primer 5′-CTAGCAGTTTGGGCTTTCCTCCTTG-3′. Sequencing was performed with the forward primer to confirm the substitution of the oligonucleotide.

### 2.6. Establishment of p53-Knockout Cells by CRISPR–Cas9

The gRNA sequence targeting the Trp53 exon7 was searched using CHOPCHOP (http://chopchop.cbu.uib.no/, accessed on 26 April 2018), and two sequences were applied; for KO1, 5′-ATAGTGGGAACCTTCTGGGACGG-3′, and for KO2, 5′-TCTGTACGGCGGTCTCTCCCAGG-3′ (the protospacer adjacent motif (PAM) is underlined). The oligos were annealed and ligated into a plentiCRISPRv2 vector (Addgene, Watertown, MA, USA) digested with *Bsm*BI according to the protocol provided by the Zhang laboratory (https://www.addgene.org/crispr/zhang/, accessed on 25 April 2018). After confirmation of the inserted sequence, the plentiCRISPRv2 vectors were co-transfected with PsPAX2 and pCMV-VSV-G vectors (Addgene) into Lenti-X 293T cells with Fugene HD to produce infectious lentivirus. The virus-containing medium was added to AX or AXT cells, and infected cells were selected with puromycin. To obtain knockout cells, puromycin-resistant cells were subjected to single-cell cloning. The expression of p53 in the clones was evaluated by Western blotting. Genomic DNA was collected from the p53 knockout clones, and the targeted region was amplified with forward primer 5′-GTCTCTTATCTGTGGCTTCTCG-3′ and reverse primer 5′-CCTTGAGGGTGAAATACTCTCC-3′. Sequencing was performed with the forward primer to examine the alteration of the sequence (data not shown).

### 2.7. Immunoblot Analysis

The cell lysate was prepared with a 2× Laemmli sample buffer (Bio-Rad, Hercules, CA, USA) supplemented with β-mercaptoethanol. Immunoblot analyses were conducted according to the standard semidry transfer procedures using 5–20% gradient precast polyacrylamide gels (ePAGEL, ATTO, Tokyo, Japan). The primary antibodies for mouse p53 purchased from Abcam (Cambridge, UK (#ab90363)) or Cell Signaling Technology (Danvers, MA, USA (#32532)) and for α-tubulin from Sigma-Aldrich (Burlington, MA, USA (#T9026)) were used.

### 2.8. Cell Cycle Analysis

The cells were trypsinized, washed with a PBS, and fixed with 70% ethanol for ≥48 h at −20 °C. Then, the cells were washed twice with ice-cold PBS and stained with a PBS containing 10 μg/ml propidium iodide and 20 μg/mL RNase. The DNA content of at least 10,000 singlet cells was analyzed by flow cytometry (FACSVerse, BD Biosciences, Franklin Lakes, NJ, USA). The data were analyzed with the FlowJo 7.6.5 (BD Biosciences).

### 2.9. Animal Care

All animal care and procedures were performed in accordance with the guidelines of Hoshi University (approval number: 29-118). The mice were housed in ventilated cages (floor area, 501 cm^2^; five mice per cage) with ALPHA-dri bedding (Shepherd Specialty Papers, Milford, NJ, USA) under specific pathogen-free conditions. The mice were fed a standard chow diet and water ad libitum and were inspected daily to ensure that they were not under distress throughout the experiments. The rooms were temperature-controlled at 22 °C and kept on a 12-h light/dark cycle.

### 2.10. Tumor Xenograft Model

The detailed schedules are shown in Figures 3c and 5a,c. To establish tumor xenografts, AX or AXT cells (including the p53-knockout AX or AXT cells) suspended in 100 μL IMDM were injected subcutaneously and bilaterally into the flanks of 12-week-old C57BL/6 SCID mice (purchased from The Jackson Laboratory, Bar Harbor, ME, USA) or 7-week-old female syngeneic C57BL/6J mice (SLC, Shizuoka, Japan), respectively, under 2% isoflurane anesthesia (Wako, Tokyo, Japan). The criteria regarding the endpoints were as follows: (1) the mean tumor diameter exceeds 20 mm; (2) the combined tumor burden exceeds 15% body weight (10-week-old mice had a body weight of ~20 g); (3) there is ulceration, infection, or necrosis of the tumor; (4) the body weight loss exceeds 20% of the baseline weight. Because of no deviation from the criteria, none of the endpoints were applied in this study. The major and minor axes of the bilateral tumors were measured, and the estimated tumor weight was calculated using the following formula, with reference to the guideline of Washington State University (https://iacuc.wsu.edu/documents/2017/12/tumor-burden-guidelines.pdf/, accessed on 22 April 2020): estimated tumor weight (mg) = tumor volume (mm^3^) = d^2^ × D/2, where d and D are the shortest and longest diameters in mm, respectively. Before analyses, the mice were euthanized with an intraperitoneal injection of a lethal dose (100 mg/kg) of pentobarbital sodium (Tokyo Kasei Kogyo, Tokyo, Japan).

### 2.11. Immunohistochemistry

Immunohistochemical analysis was performed using the standard methods [5,6,11]. Deparaffinized sections were stained with an antibody to GFP (Santa Cruz Biotechnology #sc-8334; Dallas, TX, USA). Hematoxylin was used for nuclear staining.

### 2.12. Chromatin Immunoprecipitation (ChIP)

For ChIP sequencing, AXT cells were incubated in a growth medium containing 1% formaldehyde at room temperature for 15 min. Fixation was stopped with 0.125 M glycine for 5 min, and the cells were collected by scraping from the culture surface. The cells were washed twice with a PBS, pH 7.4, containing 0.5% Igepal (Sigma-Aldrich) and 1 mM PMSF (Sigma-Aldrich), and were then snap-frozen. ChIP sequencing for p53, H3K4me3, and H3K9me3 was performed by Active Motif epigenetic services (Tokyo, Japan). Briefly, chromatin was isolated by the addition of a lysis buffer, followed by disruption with a Dounce homogenizer. The lysates were sonicated and the DNA was sheared to an average length of 300–500 bp. Genomic DNA (input) was prepared by treating aliquots of chromatin with RNase, proteinase K, and heat for decrosslinking, followed by ethanol precipitation. The pellets were resuspended, and the resulting DNA was quantified on a NanoDrop spectrophotometer. Extrapolation to the original chromatin volume allowed the total chromatin yield to be quantified. An aliquot of chromatin (30 μg for p53 and 15 μg for H3K4me3 and H3K9me3) was precleared with protein A agarose beads (Invitrogen). Genomic DNA regions of interest were isolated using a p53 antibody (Santa Cruz Biotechnology, #sc6243) or H3K4me3 or H3K9me3 (Active Motif, #39159 or #39161). The complexes were washed, eluted from the beads with an SDS buffer, and treated with RNase and proteinase K. Crosslinking was reversed by incubation overnight at 65 °C, and ChIP DNA was purified by phenol–chloroform extraction and ethanol precipitation.

For quantitative ChIP–PCR (ChIP–qPCR), ChIP was conducted as described previously [34]. AO and AXT cells were collected with or without 0.5 μM doxorubicin treatment for 8 h in triplicate, cross-linked with 1% formaldehyde for 10 minutes, and the cells were resuspended in a lysis buffer (50 mM Tris HCl, pH 8.0, 1 mM EDTA, 1% (*w*/*v*) SDS). The cell suspension was sonicated to shear DNA using a Bioruptor UCD-250 (Cosmo Bio, Tokyo, Japan). As the input DNA, 20 μL of the sheared chromatin were used. Sheared chromatin (30 μg) was incubated with 10 μL of the anti-p53 antibody (Cell Signaling Technology, #32532) at 4 °C overnight with rotation. The immune complex was collected with a Dynabeads Protein G (Veritas Corporation, Tokyo, Japan), and the collected beads were washed with a RIPA buffer (50 mM Tris HCl, pH 8.0, 1 mM EDTA, 1% (*w*/*v*) Triton X-100, 0.1% (*w*/*v*) sodium deoxycholate (DOC)) containing 150 mM NaCl twice, a RIPA buffer containing 500 mM NaCl twice, a LiCl wash buffer (10 mM Tris HCl, pH 8.0, 0.25 M LiCl, 1 mM EDTA, 0.5% (*w*/*v*) NP-40, 0.5% (*w*/*v*) DOC), and 1× TE (10 mM Tris HCl, pH 8.0, 1 mM EDTA) containing 50 mM NaCl. The beads were resuspended in 1× TE, and the crosslink was reversed with 200 mM NaCl at 65 °C overnight. After the RNase A and proteinase K treatment, DNA was recovered by phenol–chloroform extraction and ethanol precipitation and dissolved in 40 μL 1 × TE. One μL of the DNA was used for ChIP–qPCR using the primers listed in Table 2.

### 2.13. ChIP Sequencing

Illumina sequencing libraries were prepared from the ChIP and input DNAs using the standard consecutive enzymatic steps of end polishing, dA addition, and adapter ligation. After the final PCR amplification step, the resulting DNA libraries were quantified and sequenced on an Illumina NextSeq 500 (75 nt reads, single-end). The reads were aligned to the mouse genome (mm10) using the BWA algorithm (default settings). Duplicate reads were removed, and only uniquely mapped reads (mapping quality ≥ 25) were used for further analysis. Alignments were extended in silico at their 3’ ends to a length of 200 bp, which is the average length of genomic fragments in a size-selected library, and assigned to 32 nt bins along the genome. The resulting histograms (called genomic signal maps) were stored as bigWig files. The locations of peaks were determined using the MACS algorithm [35] (v2.1.0) with a cutoff of *p*-value = 1 × 10^−7^. The peaks that were on the ENCODE blacklist of known false ChIP-seq peaks were removed. Signal maps and peak locations were used as the input data for Active Motif’s proprietary analysis program, which creates Excel tables containing detailed information on sample comparisons, peak metrics, peak locations, and gene annotations. The integrative genomics viewer (IGV_2.9.2, https://software.broadinstitute.org/software/igv/, accessed on 11 March 2021) was used to visualize tracks. To compare peak metrics between two or more samples, “active regions” were defined by the start coordinate of the most upstream interval and the end coordinate of the most downstream interval (this technique is known as a union of overlapping intervals or merged peaks). The use of active regions was necessary because the locations and lengths of the intervals are rarely identical in different samples (Figure 6a). The accession number for the ChIP-seq datasets reported in this paper is GSE211492.

### 2.14. Statistical Analysis

All the assays were performed at least in triplicate. Unless indicated otherwise, quantitative data are expressed as the means ± SD or SE relative to the control value. The data were analyzed with Student’s *t*-test, and a *p*-value of < 0.05 was considered statistically significant (* *p* < 0.05; **, *p* < 0.005; NS, not significant).

## 3. Results

### 3.1. Alteration of the Mutation Status of p53 in Osteosarcoma Cells

Mouse AX cells are tumorigenic and develop into metastatic osteosarcoma in C57BL/6 mice when given by subcutaneous or intra-bone inoculation [5,9,11]. Previously, we performed exome-seq analysis in AX and AXT cells and detected an R270C missense point mutation in the p53-binding domain that corresponds to R273 in humans (Figure 1a,b).

Based on the whole genome resource (v95) of the COSMIC mutation database (https://cancer.sanger.ac.uk/cosmic, accessed on 9 May 2022), the mutation position of amino acid 273 in p53 is the most frequent hotspot in human malignancies (TP53 in humans) [36]. Amino acid substitutions R273H, R273C, and R273L are found in descending frequency in malignant tumors. Codon 273 is also one of the most common mutation sites among the six hotspots in osteosarcoma [20]. Since sequencing was performed using cDNA that was reverse-transcribed from the total RNA of AX cells, the finding that the height of the peaks of wild-type and mutant sequences was the same suggested that the mRNA expression level of wild-type and mutant p53 was also the same. Therefore, the mutation was heterozygous, implying that AX cells harbor one wild-type and one mutant allele. AXT cells, which were established from AX-derived osteosarcoma, exhibited loss of heterozygosity; that is, the wild-type allele was replaced with a mutant allele (Figure 1a).

### 3.2. The Function of Wild-Type p53 Was Preserved in the Presence of the R270C Mutant

The protein level of wild-type p53 is upregulated by its stabilization in response to DNA damage [37,38]. Treatment of AX cells with doxorubicin increased the expression level of p53, suggesting that the behavior of wild-type p53 was preserved in response to DNA damage (Figure 2a). The p53-knockout cells, which were established from AX cells using CRISPR–Cas9 and two different target sequences, showed loss of the p53 protein even when treated with doxorubicin (Figure 2a). Although the growth of the p53-knockout AX cells was still high, it tended to be lower than that of AX cells (Figure 2b). The parental AX cells were sensitive to doxorubicin. On the other hand, the p53-knockout AX cells were more resistant to doxorubicin than the parental cells, indicating that the function of wild-type p53 was maintained even in the presence of mutant p53 (Figure 2c). The transcriptional activation function of p53 was maintained in AX cells (Figure 2d,e). The mRNA expression levels of *Cdkn1a* and *Mdm2*, both of which are well-known p53 target genes [32,39], were increased by doxorubicin treatment. As expected, this transcriptional activation function was lost in the p53-knockout cells (Figure 2d,e). These findings indicated that the R270C mutation located in the DNA-binding domain did not suppress the transcriptional activation function of wild-type p53, including that related to the DNA damage response.

### 3.3. The Simultaneous Presence of Wild-Type and Mutant p53 Does Not Affect the Tumorigenic Activity In Vivo

We further examined how the phenotype of AX cells changes when p53 is knocked out. Previous reports indicated that the human R273C p53 mutant enhances the invasion ability of several cancer cell lines [40]. Both the parental AX cells and the p53-knockout cells invaded through Matrigel-coated Transwells at the same rate, and the loss of p53 expression did not affect the invasive ability of the cells (Figure 3a,b). Next, the influence of p53 expression on the tumorigenic activity in vivo was examined. In this study, we used SCID mice so that the formed tumors were more likely to be of uniform size (Figure 3c). Inoculation of mice with AX cells or the p53-knockout cells caused the formation of similarly developed tumors, although those derived from the p53-knockout cells tended to be slightly smaller (Figure 3d). In addition, tumors derived from AX cells or the p53-knockout cells exhibited the same histological findings, including osteoid and bone formation (data not shown). Given that AX cells express GFP, GFP expression using total RNA extracted from blood or tissues can be used to quantify circulating tumor cells or metastatic lesions, respectively [41]. Although there were quantitative variations, knockout of p53 did not significantly affect the amount of circulating tumor cells or lung metastasis (Figure 3e,f) These findings suggest that the simultaneous presence of the wild-type p53 allele and the R270C mutation did not alter tumor initiation or the progression of osteosarcoma in vivo.

### 3.4. The Functions of Mutant p53 Overexpressed in AXT Cells

In AXT cells, both wild-type p53 alleles were replaced with R270C mutant p53 and exhibited loss of heterozygosity (Figure 1a). Next, we investigated the biological effects of mutant p53. The expression of Trp53 mRNA was higher in AXT cells than in AX cells (Figure 4a). Overexpression of the mutant p53 protein in AXT cells is likely due to the longer half-life of mutant p53; this phenomenon is observed in human cancers [42] (Figure 4b). We used CRISPR–Cas9 to deplete mutant p53 in AXT cells and establish knockout cells (Figure 4b). The proliferation rate of the knockout cells was high but lower than that of the parental AXT cells, suggesting that R270C mutant p53 might enhance cell growth as previously reported regarding other types of mutant p53 [33] (Figure 4c). The cell cycle status showed that the S-phase fraction was lower in the knockout cells than in AXT cells (Figure 4d,e). Apoptotic cells, evaluated by the sub-G1 fraction (Figure 4d,e) and the cell morphology, were not increased by the knockout of mutant p53. AXT cells lacking wild-type p53 were more resistant to doxorubicin than AX cells, and the level of resistance was the same as that in the p53-knockout AX cells (Figure 2c and Figure 4f). Depletion of mutant p53 did not affect the sensitivity to doxorubicin (Figure 4f); this finding is consistent with our findings below showing that mutant p53 loses its transcriptional activation function. Notably, the invasion ability of AXT cells was not affected by the depletion of mutant p53 (Figure 4g,h), suggesting that the R270C p53 mutation enhances proliferation but does not play a significant role in the invasive properties of AXT cells.

### 3.5. Loss of Mutant p53 Did Not Prevent Metastatic Progression In Vivo

AXT cells are highly tumorigenic in syngeneic C57BL/6 mice [6,7,8,9,10,41]. We examined the role of mutant p53 in tumorigenesis and tumor progression and investigated whether mutant p53 could be a potential target for osteosarcoma. The growth of primary tumors derived from the p53-knockout AXT cells was slower than that of the tumors derived from the parental AXT cells (Figure 5a,b). This finding might be due to the differences in the growth of the two cell types, as shown in our in vitro experiments (Figure 4c). Then, to evaluate the metastatic ability of primary tumors of equal size, the analyses were performed at different times (Figure 5c,d). All the tumors were histologically the same, including the presence of intratumor osteoid and bone formation (Figure 5e). Lung metastatic lesions were histologically detected by GFP staining (Figure 5f). The size of lung metastases in the mice inoculated with AXT cells tended to be larger than that in the mice with the p53-knockout cells, although the difference was not statistically significant (Figure 5g). RT-qPCR analyses using the GFP expression level showed that there was no significant difference in the amount of circulating tumor cells or lung metastases between the parental AXT cells and the p53-knockout cells (Figure 5h,i).

Collectively, these results show that in the absence of R270C mutant p53, primary and metastatic lesions could be generated, and tumor progression in vivo continued.

### 3.6. R270C Binds to the Vicinity of the Transcription Start Sites of Multiple Genes

Our results thus far show that R270C mutant p53 did not have a critical role in the progression of osteosarcoma in vivo. Next, we used ChIP-seq analysis to examine whether mutant p53 could bind to chromatin in AXT cells. ChIP-seq analysis of H3K4me3 and H3K9me3 was performed as a marker of transcriptional activation and repression, respectively. Our results indicated that R270C mutant p53 bound to chromatin; 8277 peaks were detected. Many mutant p53-binding regions overlapped with those of H3K4me3, indicating that R270C bound to many genes that were ready to be transcriptionally activated (Figure 6a). The binding regions that overlapped between mutant p53 and H3K9me3 were also detected, albeit much less frequently. Notably, R270C mutant p53 bound in the vicinity of the transcription start sites of *Cdkn1a* and *Mdm2* (Figure 6b,c). In addition, the R270C mutant bound to the transcription start sites of *Ccnd1* and *Trp53* (Figure 6d,e), both of which are highly expressed in AXT cells [41] (Figure 4a,b). However, unlike the findings in the AX cells (which contain wild-type p53), treatment of AXT cells with doxorubicin did not increase the expression of *Cdkn1a* or *Mdm2* (Figure 6f,g). These findings indicate that binding of R270C mutant p53 to the transcription start sites of many genes does not correlate with the regulation of gene expression or necessarily inhibit the transcription of those genes.

### 3.7. R270C Mutant p53 Exhibits a Different Binding Profile from Wild-Type p53

To reveal the molecular aspects of discrepancy between DNA binding and loss of transcriptional activation function in R270C mutant p53, its binding was compared to that of wild-type p53 by ChIP-qPCR analyses. AO cells were mouse osteosarcoma cells established from bone marrow stromal cells derived from *Ink4a/Arf*-null mice like AX cells but exhibited a lower tumorigenic activity as well as a different differentiation capability [5,43]. AO cells harbored wild-type p53 alleles (Figure 7a) and the mRNA expression levels of *Cdkn1a* and *Mdm2* were upregulated after the doxorubicin treatment (Figure 7b). The expression level of the p53 protein was increased by the doxorubicin treatment in AO cells but, notably, the amount of the p53 protein was more abundant in AXT cells compared to AO cells (Figure 7c).

In human *CDKN1A*, the p53-responsive element, which was reported to possess enhancer activity, is located at the over 2000 bp upstream region of the transcription start site [44,45,46,47]. Therefore, we analyzed the binding levels of wild-type p53 at the upstream regions of the transcription start site in the mouse AO cells (Figure 7d). Consistent with the human data, higher levels of wild-type p53 binding were detected at the around 1927 bp upstream region in the mouse AO cells, and the binding levels further increased after the doxorubicin treatment (Figure 7e). In contrast, the increase in mutant p53 binding at this region after the doxorubicin treatment was much smaller in AXT cells (Figure 7f). As ChIP-seq analysis suggests (Figure 6b and Figure 7d), the mutant p53 binding level around the transcription start site was basically high in AXT cells. The wild-type p53 binding level in AO cells in this region was lower as previously reported in human p53 [46]. In addition, higher mutant p53 binding was also detected in the +6207 downstream region after the doxorubicin treatment, while wild-type p53 binding was not detected in this region.

Thus, R270C mutant p53 exhibited a different binding profile from wild-type p53, and the binding levels at the genomic region critical for transcriptional activation, namely the upstream p53-responsive element, were much lower than those of wild-type p53.

## 4. Discussion

In this study, we examined the tumorigenic activity of the R270C form of mutant p53 in vivo and determined its metastatic potential in osteosarcoma cells. The R270C mutant is equivalent to the human R273C mutant in the DNA-binding domain (Figure 1b) that occurs in about 2.7% of all human malignancies, including osteosarcoma, and is one of the frequently substituted “hotspot” amino acids [14,18,20]. AX cells express similar levels of both wild-type and mutant p53 (Figure 1a). Notably, treatment of AX cells with doxorubicin upregulated the expression of the downstream targets of p53 (Figure 2d,e). This finding suggests that the R270C mutation did not hinder the normal function of wild-type p53 even though the transcriptional activation function of the R270C mutant was completely lost (Figure 6f,g). Moreover, AX cells that were p53-heterozygous or p53-null showed a differential sensitivity to doxorubicin, but this difference in sensitivity did not correlate with the metastatic potential in vivo (Figure 3c–f).

Rather, the p53 mutation might influence the initiation or development of osteosarcoma but not the maintenance of this type of tumor. Consistent with this notion, previous reports demonstrate that deletions and mutations in p53 promote osteosarcoma development in mice [23,24,25,26,27,28]. In our osteosarcoma model, serial transplantation of AX cells leads to the loss of heterozygosity of p53 (Figure 1a). A recent study suggested that the human R273C mutant can modify the function of wild-type p53 in prostate cancer cells and alter downstream transcriptional networks [48]. These mechanistic roles of mutant p53 might promote the initiation of osteosarcoma.

Intriguingly, ChIP-seq analysis of mutant p53 in AXT cells revealed that the R270C mutant can bind to chromatin in the vicinity of the transcription start sites of various genes, including well-known p53 target genes as well as highly expressed genes such as *Ccnd1* and *Trp53* (Figure 6). In addition, many of the p53-binding regions overlapped with those of transcriptional active marker H3K4me3 (Figure 6a). However, transcription activation of the target gene was not observed after the doxorubicin treatment. These findings suggest that even though the R270C mutant binds around the transcription start sites of the p53 target genes, the mutant cannot activate their transcription. The levels of binding of mutant p53 to the critical regulatory region, such as the p53-responsive element, was lower despite expressing a large amount of the protein compared to wild-type p53 (Figure 7f). Conversely, mutant p53 binds to the genomic regions where wild-type p53 does not bind. This alteration of the binding profile is a potential mechanism of the loss of transcriptional activation function. Previous reports using a ChIP assay demonstrated that the human R273C mutant bound to the HER2 promoter regions [49], while R273C that is transduced into p53-knockout LNCaP cells (generated with CRISPR) cannot bind chromatin [48]. Thus, the ability of the R273C mutant to bind chromatin might depend on the cell context or how p53 is modified to produce the mutant form.

Previous reports indicated that R273C p53 has a gain of function. Overexpression of R273C in Saos2 cells upregulates the transcriptional activity of *HER2* [49]. R273C was also suggested to regulate the expression of *AXL* through histone acetylation in lung cancer cells [50]. In addition, in models of lung and breast cancer, the R273C mutation confers cancer cells with a malignant phenotype; enhances colony formation, invasion ability, and/or drug resistance in vitro; and enhances tumorigenic growth in vivo [51]. Very recently, it was shown that R273C expressed in prostate cancer cells cannot independently bind to DNA and loses its transcriptional transactivation activity; however, modification of wild-type p53 results in enhanced colony formation, increases tumor cell survival after irradiation in vitro, and alters tumorigenic activity in vivo [48]. Our results using AXT cells also suggest that R270C mutant p53 is involved in the enhancement of cell growth (Figure 4c–e). However, the knockout of the R270C mutant did not prevent the invasion ability of cells or the in vivo progression of osteosarcoma, including lung metastasis (Figure 4 and Figure 5). Histological differences were not observed between the tumors derived from the parental cells and the knockout AXT cells (Figure 5e). Consistent with our results, a study of Hep3B human hepatocarcinoma cells suggested that the expression of the R248Q form of mutant p53 produced phenotypes that were different, such as drug resistance, from those induced by the expression of the R273C mutant [52]. Therefore, the aberrant function of mutant R273C might depend on the cell context and genetic background of the cell, such as the status of activating oncogenes.

Importantly, whether mutant p53 can be used as a clinical prognostic marker in osteosarcoma patients remains controversial [53]. Studies using clinical samples indicate that alterations in p53 correlate with markedly reduced event-free survival [54]. Three osteosarcoma patients who carried the R337H p53 mutation had a poorer outcome than other patients [55]. Mutations in p53 are suggested to be closely associated with osteosarcoma progression [56]. A large-scale study or meta-analysis showed that TP53 gene alterations correlate with decreased survival [33,57]. In contrast, an analysis of 196 patients with high-grade osteosarcoma demonstrated that the prognosis of patients with p53 mutations is only slightly worse than those with the wild-type gene; a relationship between the p53 status and systemic relapse was not identified, and a p53 mutation did not predict the development of metastasis [17].

Finally, accumulating evidence indicates that mutant p53 is a possible therapeutic target for several types of malignancies including osteosarcoma [29,30,31,32,33]. Given that the role of mutant p53 in the progression of osteosarcoma in vivo has not been sufficiently analyzed, we reason that it remains to be elucidated whether mutant p53 is a suitable therapeutic target for osteosarcoma. Our study suggested that the loss of the R270C mutant did not prevent osteosarcoma metastatic progression in vivo; therefore, the therapeutic potential targeting the R270C mutant (equivalent to human R273C) is limited (Figure 3 and Figure 5). Osteosarcoma consists of heterogeneous populations of disease phenotypes, and the oncogenes that drive this tumor remain unclear [1]. Consistent with the heterogeneous nature of osteosarcoma, it is likely that the disease phenotypes induced by the status of p53 and activated oncogenic pathways differ depending on the cell context.

Overall, given that recent advances in technology have resulted in the more frequent detection of p53 mutations, it is important to examine the biological roles of mutant p53 using preclinical models and reevaluate the clinical significance of p53 alterations in response to chemotherapy and metastasis in human osteosarcoma.

## Figures and Tables

**Figure 1 cells-11-03614-f001:**
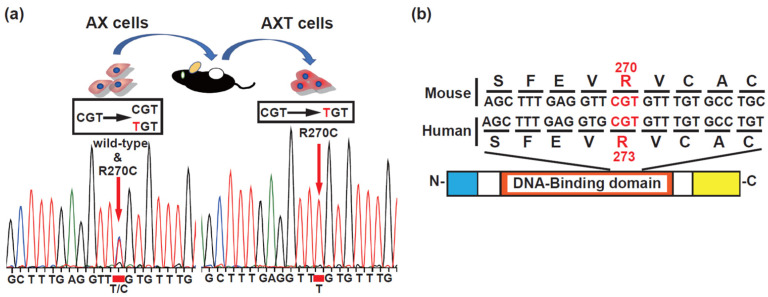
AX and AXT cells contain R270C mutant p53. (**a**) AX and AXT cells contain a missense mutation in the DNA-binding domain that occurs as a result of amino acid substitution. (**b**) A comparison of mouse *Trp53* and human *TP53* sequences at the mutation site. The red text indicates the mutated codon.

**Figure 2 cells-11-03614-f002:**
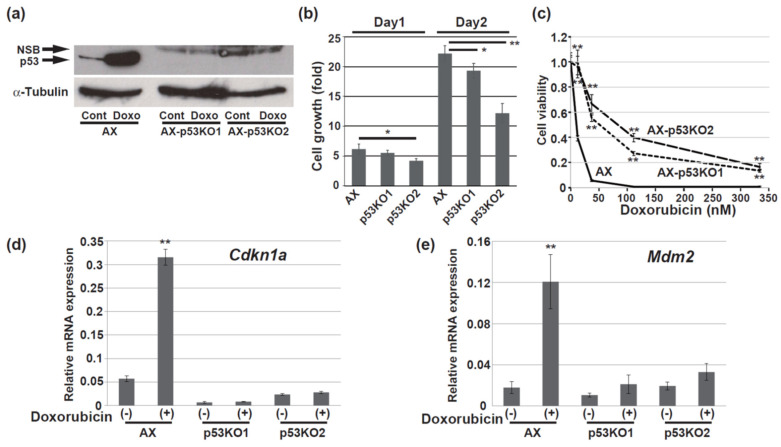
The R270C mutant does not block the function of wild-type p53. (**a**) Immunoblot analysis of p53 expression using an antibody (#ab90363) in the AX and p53-knockout AX cells with or without 0.5 µM doxorubicin for 16 h. KO, knockout; NSB, nonspecific bands; Doxo, doxorubicin. (**b**) The growth of the AX and p53-knockout AX cells. The ratio relative to the value for day 0 was calculated for each datapoint. (**c**) The viability of the AX and p53-knockout AX cells was assessed after 2-day exposure to the indicated concentrations of doxorubicin. The statistical significance between AX cells and the respective knockout cells is shown. (**d**,**e**) RT and real-time PCR analysis of *Cdkn1a* (**d**) and *Mdm2* (**e**) mRNA in the AX and p53-knockout AX cells treated with or without 0.5 µM doxorubicin for 16 h. The data are normalized to the corresponding levels of *Actb* mRNA and are shown as the means ± SD of triplicate values. (* *p* < 0.05; ** *p* < 0.005).

**Figure 3 cells-11-03614-f003:**
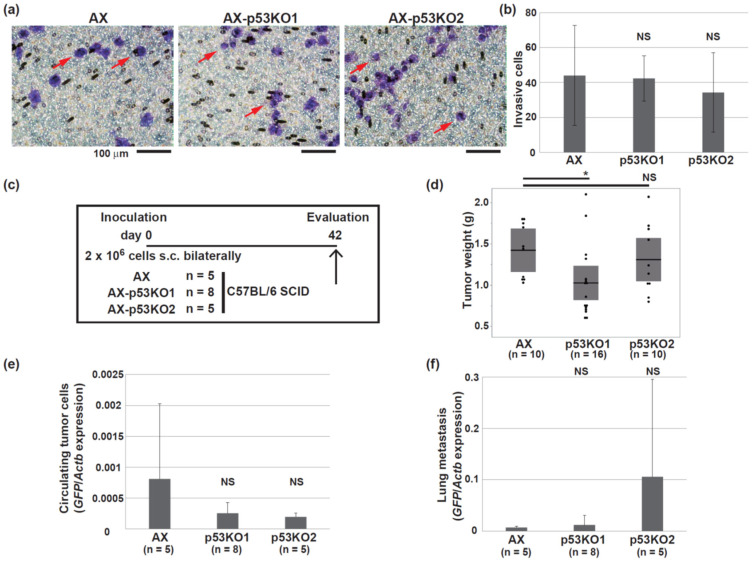
Knockout of wild-type and mutant p53 in AX cells did not prevent tumor progression. (**a**,**b**) Microscope images of the invasion assay using the AX and p53-knockout AX cells. Representative purple-stained cells are indicated with arrows. The mean number of invasive cells counted in three different fields of each transwell in triplicate experiments is shown in (**b**). (**c**) The schedule of cell inoculation into C57BL/6 SCID mice. (**d**) The dot plot of weight of primary tumors. The horizontal bars and boxed regions show the means and the 95% confidence intervals, respectively. (**e**,**f**) RT and real-time PCR analysis of *GFP* mRNA in whole blood (**e**) or a lung (**f**) derived from mice inoculated with the indicated cells. (* *p* < 0.05; NS, not significant).

**Figure 4 cells-11-03614-f004:**
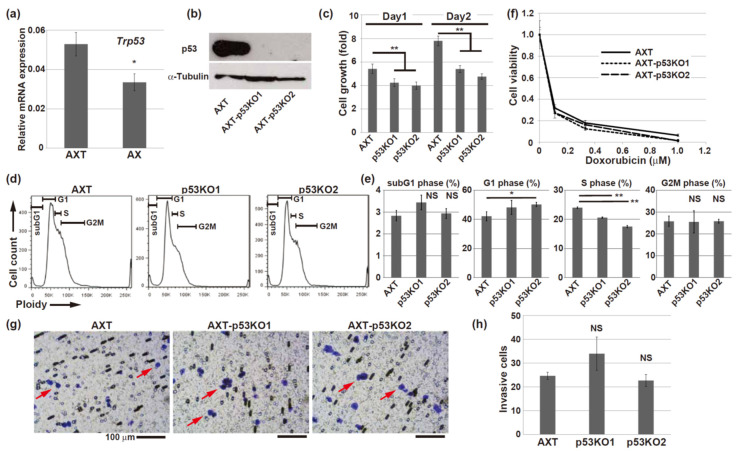
Knockout of R270C mutant p53 did not affect the invasion ability of AXT cells. (**a**) RT and real-time PCR analysis of *Trp53* mRNA in AX and AXT cells. The data are normalized to the corresponding levels of *Actb* mRNA and are shown as the means ± SD of triplicate values. (**b**) Immunoblot analysis of p53 expression using an antibody (#ab90363) in the AXT and p53-knockout AXT cells. (**c**) The growth of the AXT and p53-knockout AXT cells. The ratio relative to the value for day 0 was calculated for each datapoint. (**d**,**e**) Flow cytometry analysis of DNA content in the AXT and p53-knockout AXT cells. The size of each fraction of cells is shown in (**e**). (**f**) The viability of the AXT and p53-knockout AXT cells was assessed after 2-day exposure to the indicated concentrations of doxorubicin. (**g**,**h**) Microscope images of the invasion assay using the AXT and p53-knockout AXT cells. Representative purple-stained cells are indicated with arrows. The mean number of invasive cells counted in three different fields of each transwell in triplicate experiments is shown in (**h**). (* *p* < 0.05; ** *p* < 0.005; NS, not significant).

**Figure 5 cells-11-03614-f005:**
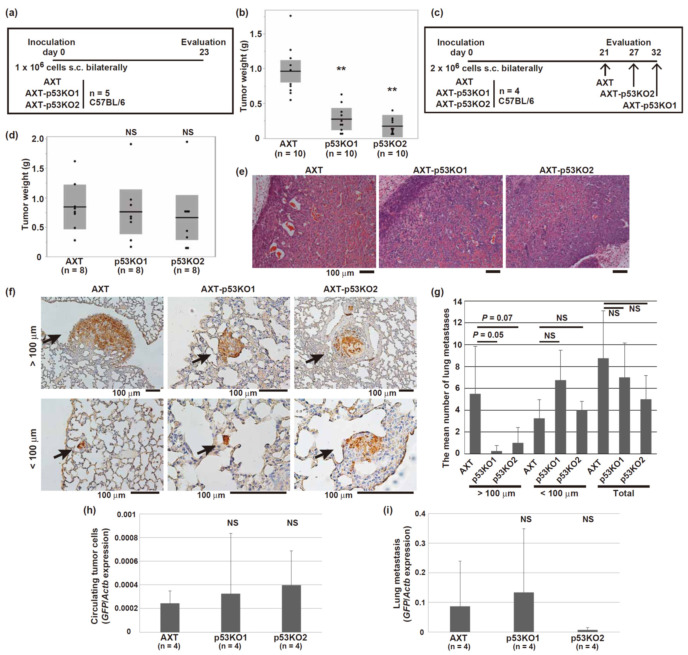
Depletion of mutant p53 did not prevent osteosarcoma progression in vivo. (**a**,**c**) The schedule of cell inoculation into syngeneic C57BL/6 mice. (**b**,**d**) The dot plot of weight of primary tumors. The horizontal bars and boxed regions show the means and the 95% confidence intervals, respectively. (**e**) Hematoxylin-eosin (H&E) staining of primary tumors in the mice inoculated with the indicated cells. (**f**) Representative images of the lung metastatic lesions stained with GFP. The arrows indicate lung metastases classified according to whether the major axis was larger than 100 μm. (**g**) The number of metastases present throughout a randomly sliced section was counted. The data represent the mean values from the right lungs of four mice. (**h**,**i**) RT and real-time PCR analysis of *GFP* mRNA in whole blood (**h**) or a lung (**i**) derived from the mice inoculated with the indicated cells. (** *p* < 0.005; NS, not significant).

**Figure 6 cells-11-03614-f006:**
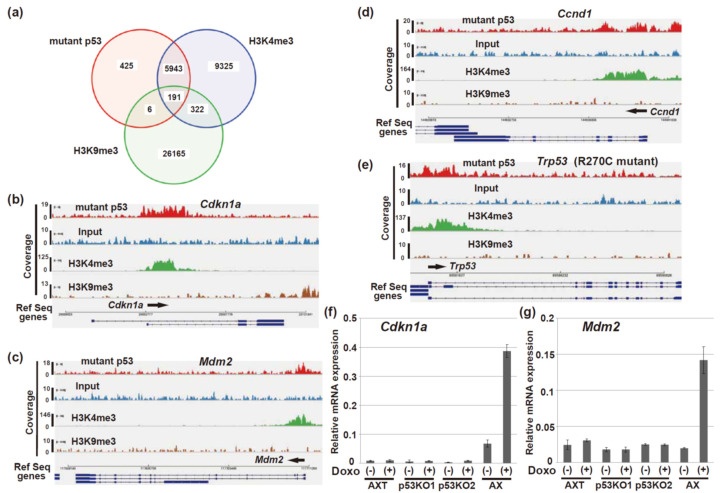
R270C mutant p53 in AXT cells binds to chromatin. (**a**) A Venn diagram illustrating the genomic overlap of peaks in the active regions of p53, H3K4me3, and H3K9me3. The active regions are defined in the Methods Section. (**b**,**c**) Density plots of the ChIP-seq reads for p53, H3K4me3, and H3K9me3 at the *Cdkn1a* (**b**) or *Mdm2* (**c**) loci. The data are based on the snapshot images taken from analysis using the IGV software. (**d**,**e**) ChIP-seq occupancy profiles for p53, H3K4me3, and H3K9me3 at the *Ccnd1* (**d**) and *Trp53* (**e**) loci. (**f**,**g**) RT and real-time PCR analysis of the *Cdkn1a* (**f**) and *Mdm2* (**g**) mRNA in the AX, AXT, and p53-knockout AXT cells treated with or without 0.5 µM doxorubicin for 16 h. The data are normalized to the corresponding levels of *Actb* mRNA and are shown as the means ± SD of triplicate values.

**Figure 7 cells-11-03614-f007:**
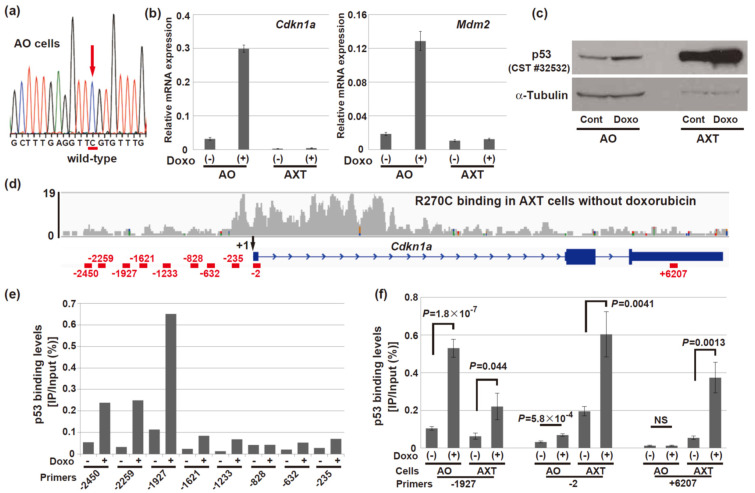
R270C mutant p53 exhibits a different DNA-binding profile from wild-type p53. (**a**) AO cells harbor wild-type p53. (**b**) RT and real-time PCR analysis of the *Cdkn1a* and *Mdm2* mRNA in AO and AXT cells treated with or without 0.5 µM doxorubicin for 8 h. The data are normalized to the corresponding levels of *Actb* mRNA and are shown as the means ± SD of triplicate values. (**c**) Immunoblot analysis of p53 expression in AO and AXT cells with or without 0.5 µM doxorubicin for 16 h. (**d**) Map of the *Cdkn1a* gene locus showing the binding profile of R270C mutant p53 in AXT cells, the transcription start site (+1), and the location of ten amplicons used in real-time qPCR analyses. Numbers indicate the position of the 5’ end of the amplicon relative to the transcription start site. The snapshot image was taken from analysis using the IGV software. (**e**) Screening of the p53 binding levels of wild-type p53 in AO cells treated with or without 0.5 μM of doxorubicin for 8 h. The result of a single experiment is shown. (**f**) DNA binding levels of wild-type p53 or R270C mutant p53 were evaluated by ChIP-qPCR analyses. ChIP assays were performed with cell extracts obtained from AO or AXT cells treated with or without 0.5 μM doxorubicin for 8 h. The results shown are the means ± SE of nine independent PCRs of three independent experiments. (NS, not significant).

**Table 1 cells-11-03614-t001:** Sequences of primers and predicted product sizes for real-time RT–PCR analysis.

Gene Symbol	Forward	Reverse	Product Size (bp)
*GFP*	GACGTAAACGGCCACAAGTT	TTGCCGGTGGTGCAGATGAA	95
Mouse genes			
*Trp53*	ACTTACCAGGGCAACTATGG	CTGGCAGAATAGCTTATTGAGG	105
*Cdkn1a*	CAAAGTGTGCCGTTGTCTCTTC	GTCAAAGTTCCACCGTTCTC	112
*Mdm2*	CTAGACTGTCTACCTCATCTAG	CAGGCTCGGATCAAAGGACA	109
*Actb*	CAACCGTGAAAAGATGACCC	TACGACCAGAGGCATACAG	102

**Table 2 cells-11-03614-t002:** Sequences of primers and predicted product sizes for ChIP–qPCR analysis.

Primer Name	Forward	Reverse	Product Size (bp)
−2450	GAAAGACTGAGTAGTCCCAGAC	GTGCCTTTACCCTACTGGTG	114
−2259	GTCACTTCTATCTGAGAAGC	CATCCAAGTCGTCCATCCCA	102
−1927	CGATCTCTAGACATCGGAGA	CAGAGACTGGAGTCTTAGTTTG	107
−1621	GGCAAGCGCTATATTAACGGAG	CTGAAATCACGGTACTTGGG	115
−1233	GTCTTACTGCTATGTCTGTC	GGGAAATGTCTAATACTCCC	114
−828	CTGTGAGACAGGGAGGAAATG	AAAATCCCAAGAAGTCCCAC	106
−632	GTGCCTCAATCTCCCAAGTA	CTATTCCGATGGAGACCAAC	114
−235	CATAGATGTATGTGGCTCTG	AATCTAAGCCCGCGCCAGACA	121
−2	AACTGCAGCAGCCGAGAGGT	AAGCTCTCACCTCTGAATGTC	106
+6207	TGAAGACAGGAATGGTCCCC	GCAGCAGATCACCAGATTAAC	119

## Data Availability

Not applicable.

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
