# Peer review of "Depletion of R270C Mutant p53 in Osteosarcoma Attenuates Cell Growth but Does Not Prevent Invasion and Metastasis In Vivo"

_cells, 2022, doi:10.3390/cells11223614_

Round 1
Reviewer 1 Report
The manuscript studies the role of R270C mutant p53 in osteosarcoma. The study revealed that knockout of R270C mutant p53 can inhibit the growth of osteosarcoma, but has no effect on the invasion and metastasis of osteosarcoma. The significance of using R270C mutant p53 as a therapeutic target for osteosarcoma is not obvious. However, osteosarcoma is a highly heterogeneous tumor, and mutant p53 plays different roles in different tumors or cells with different genetic backgrounds, so this study provides a new understanding of the role of R270C mutant p53 in osteosarcoma. Overall, the manuscript is well written and the interpretation of data is generally sound. Here are some suggestions:
1. In figure 1d and figure 3c, if the growth of cells is not continuously detected, it is better to show it in the form of histogram.
2. In figure 1d, 1e, and figure 3c, significant differences between groups should be noted.
Author Response
Response to Reviewer 1 Comments
We thank the reviewer for appreciating our study.
Point 1: In figure 1d and figure 3c, if the growth of cells is not continuously detected, it is better to show it in the form of histogram.
We appreciate the reviewer’s suggestion. We have rewritten Figures 1d and 3c.
Point 2: In figure 1d, 1e, and figure 3c, significant differences between groups should be noted.
We evaluated the statistical significance and added it to Figures 1d, 1e, and 3c.
Reviewer 2 Report
Provide statistics (p-value) for 1D, 3C
Two knockouts are not similar in Fig. 1d, 2d, 3e. It’s hard to make conclusions with variable data.
Figure 4: Why is tumor weight n=8 and others n=4?
Figure 4F should be quantified for the whole lung (# of mets, size of mets, etc).
Figure 5: p53 does not look enriched at the Pparg promoter. The small increase matches the input.
You should compare ChIP of WT to mutant p53 and with and without Dox. There has to be a comparator because things are rarely “all-or-nothing”. You can’t see increases or decreases in binding. This is critical given that R273C binds less to DNA according to McMann, et al.
If the mutant p53 binds but there is no transcriptional activation, what is the mechanism?
Supplemental data not included.
ChIP-seq data should be made available.
Minor:
1. Line 178: endpoint is when it is not those criteria; i.e. if it did exceed the size.
2. Provide GFP antibody clone #.
3. Supp. Fig. S1 should be a main figure if it is supporting the main (in the title) conclusion that the knockout is slower growing (but I can’t see the supplemental data).
Author Response
Response to Reviewer 2 Comments
Point 1: Provide statistics (p-value) for 1D, 3C
We evaluated the statistical significance and added it to Figures 1d, and 3c. According to the kind suggestion of reviewer #1, we have rewritten those figures.
Point 2: Two knockouts are not similar in Fig. 1d, 2d, 3e. It’s hard to make conclusions with variable data.
We agree with the reviewer’s comment. As the reviewer pointed out, the growth rates are different between two knockout AX cells (Fig. 1d). In addition, the growth inhibitory effect by p53 knockout is very slight in AX cells in vivo (Fig. 2d). Therefore, we cannot conclude that loss of mutant p53 suppresses tumor growth from the results of AX cells alone. The reason for this is presumably because the expression level of p53 protein in AX cells is originally low under the unstimulated state. In addition, since AX cells express both wild-type and mutant p53, it is hard to purely evaluate the effect of loss of mutant p53.
On the other hand, AXT cells express a large amount of mutant p53 protein, and loss of function of mutant p53 can be evaluated purely. The significant growth inhibition was consistently observed in vitro and in vivo by the knockout of mutant p53, although there was some variability. There is no significant difference in the increase in the G1 phase with AXT-p53KO1 (Fig. 3e), but more importantly, a significant decrease in the S phase was observed in both knockout cells.
The description was rewritten taking into account the above considerations (page 8, lines 315-317 and page 9, line 348).
Point 3: Figure 4: Why is tumor weight n=8 and others n=4?
To establish tumor xenografts, cells were inoculated subcutaneously and bilaterally into mice. Therefore, the number of tumors is twice the number of mice. We added ‘bilaterally’ in Fig. 2c, 4a, and 4c.
Point 4: Figure 4F should be quantified for the whole lung (# of mets, size of mets, etc).
We appreciate the reviewer’s the constructive comment. In this study, the left lungs were used for RNA extraction to quantitate the metastatic amount by RT-qPCR, and the right lungs were subjected to immunohistochemistry. We counted the number of metastases present throughout a randomly sliced section, and classified them according to whether the major axis was larger than 100 mm. Data were shown as mean values from the right lungs of four mice. We incorporated the graph as Fig. 4g and described the findings in the manuscript in page 9, lines 368-370.
Point 5: Figure 5: p53 does not look enriched at the Pparg promoter. The small increase matches the input.
We appreciate the reviewer’s insightful observation. p53 enrichment certainly seems to match the input peak and it is unclear whether the peak indicates the genuine binding. Since no other initiation site where H3K9me3 and mutant p53 clearly bind together has been found so far, Pparg was replaced with Trp53, which is highly expressed despite mutant p53 binding like Ccnd1. (Fig. 5g, page 10, lines 401-403).
Point 6: You should compare ChIP of WT to mutant p53 and with and without Dox. There has to be a comparator because things are rarely “all-or-nothing”. You can’t see increases or decreases in binding. This is critical given that R273C binds less to DNA according to McMann, et al.
We recognize that the experiments the reviewer points out are very important for elucidating the molecular mechanisms involved in the mutant p53 phenotype demonstrated in Fg. 3 and Fig. 4. We have attempted to establish cells that stably express wild-type p53 from AXT cells, but failed so far, probably because overexpressing wild-type p53 induced cell death. Then we are trying to create a conditional expression system including the regulation of the expression level of wild-type p53 using Tet-on, and we will report on the comparison of the DNA-binding modes of wild-type and mutant p53 in the next study.
Point 7: If the mutant p53 binds but there is no transcriptional activation, what is the mechanism?
We appreciate the reviewer’s insightful comment. We speculate the possible mechanisms including: 1) The binding motif of mutant p53 is different from wild-type p53; 2) The mutant form is unable to construct a normal transcriptional machinery; 3) The amount of binding is not sufficient for transcription, which is pointed out by the reviewer. We would like to clarify the molecular mechanisms in the next study. We mentioned our hypothesis in discussion (page 12, lines 445-450).
Point 8: Supplemental data not included.
We apologize for missing the supplementary data. According to the reviewer’s insightful suggestions, Fig. S1 was incorporated into the main Fig. 4. Table S1 was also incorporated into method as Table 1.
Point 9: ChIP-seq data should be made available.
We appreciate the reviewer’s constructive suggestion. We registered the ChIP-seq data to GEO in the hope that they would be useful to researchers. The accession number for the ChIP-seq datasets reported in this paper is GSE211492. We mentioned it in the method section, page 5, lines 238-239.
Minor:
Point 1: Line 178: endpoint is when it is not those criteria; i.e. if it did exceed the size.
We thank the reviewer for the important comment. In this study none of the endpoint criteria was applied because of no deviation from the criteria. We mentioned it in page 4, lines 178 and 182-183.
Point 2: Provide GFP antibody clone #.
We used the antibody # sc-8334 to detect GFP expression and mentioned it in page 4, line 193.
Point 3: Supp. Fig. S1 should be a main figure if it is supporting the main (in the title) conclusion that the knockout is slower growing (but I can’t see the supplemental data).
We appreciate the reviewer’s suggestion. We incorporated Fig. S1 as the main figure in Fig. 4.
Round 2
Reviewer 2 Report
The differences between the two KO lines is still a problem. Which one is to be believed? Between this and the large error bars throughout the manuscript, the rigor is very low.
Point 6 is not adequately addressed, which is a big problem as this is not properly controlled to make the main point in your manuscript. In response to point 7, you hypothesize that WT is different than mutant, but you cannot justify this as you don’t have WT. Furthermore, you are stating in your manuscript that the binding sites are the same.
Point # 10 (Is there a change in apoptosis in the knockouts?) was completely ignored.
Minor point 1: You state that “The endpoint is when it did not exceed 20 mm”. Therefore, you would have had to euthanize immediately. It’s confusing. Please reword.
Author Response
Response to Reviewer 2 Comments
Point 1: The differences between the two KO lines is still a problem. Which one is to be believed? Between this and the large error bars throughout the manuscript, the rigor is very low.
We agree with the reviewer’s concerns. We observed that the two knockout cells from AX or AXT cells formed primary and metastatic lesions in vivo similar to the parental cells. Therefore, in this study, we would like to mainly consider that depletion of mutant p53 did not significantly affect the tumor progression in osteosarcoma, that is, the role of mutant p53 in this malignancy is limited. On the other hand, the two knockout cells are single cell clones established using the different target sequences of CRISPR-Cas9 and are not exactly the same cells. Although the levels of change varied between the two knockout cells, we described a decrease in doxorubicin sensitivity and in cell proliferation as the biological tendency common to the two knockout cells. Based on this notion, we rewrote the description in the revised manuscript (page 7, lines 297-298, page 8, lines 328-330 and 334-336).
Point 2: Point 6 is not adequately addressed, which is a big problem as this is not properly controlled to make the main point in your manuscript. In response to point 7, you hypothesize that WT is different than mutant, but you cannot justify this as you don’t have WT. Furthermore, you are stating in your manuscript that the binding sites are the same.
We appreciate and agree with the reviewer’s constructive comment. According to the comment, we performed ChIP-qPCR analyses and compared the DNA binding profiles between wild-type 53 and R270C mutant p53 with or without doxorubicin treatment. The results showed higher levels of wild-type p53 binding were detected at around the upstream regulatory region (p53-responsive element) of Cdkn1a in mouse AO cells which harbor wild-type p53, and the binding levels further increased after doxorubicin treatment. In contrast, the increase of R270C mutant p53 binding at this region after doxorubicin treatment was much smaller in AXT cells. Furthermore, mutant p53, but not wild-type p53, bound to the transcription start site and downstream region of Cdkn1a, suggesting that the binding profiles of mutant p53 altered.
We added these results in the revised manuscript (page 5, lines 219-236, pages 12-13, lines 436-477, Fig. 7, abstract and discussion page 13, lines 499-511).
Point 3: Point # 10 (Is there a change in apoptosis in the knockouts?) was completely ignored.
According to the comment, we evaluated apoptotic cells based on sub-G1 fraction and the cell morphology. However, we did not observe the induction of apoptosis by the knockout of mutant p53. We added this result in the revised manuscript (Fig. 4d, e and page 9, lines 358-360).
Point 4: Minor point 1: You state that “The endpoint is when it did not exceed 20 mm”. Therefore, you would have had to euthanize immediately. It’s confusing. Please reword.
We have rewritten the description regarding the endpoint properly to avoid confusion (page 4, lines 182-186).
